# Source Information Affects Interpretations of the News across Multiple Age Groups in the United States

**Robert B. Michael** [1,*] and **Mevagh Sanson** [2]

1    Psychology Department, University of Louisiana at Lafayette, Lafayette, LA 70504-3644, USA
2    School of Psychology, The University of Waikato, Hamilton 3240, New Zealand;
     mevagh.sanson@waikato.ac.nz
*    Correspondence: robert.michael@louisiana.edu; Tel.: +1-(337)-482-6585

**Abstract:** People have access to more news from more sources than ever before. At the same time, they increasingly distrust traditional media and are exposed to more misinformation. To help people better distinguish real news from "fake news," we must first understand how they judge whether news is real or fake. One possibility is that people adopt a relatively effortful, analytic approach, judging news based on its content. However, another possibility—consistent with psychological research—is that people adopt a relatively effortless, heuristic approach, drawing on cues outside of news content. One such cue is where the news comes from: its source. Beliefs about news sources depend on people's political affiliation, with U.S. liberals tending to trust sources that conservatives distrust, and vice versa. Therefore, if people take this heuristic approach, then judgments of news from different sources should depend on political affiliation and lead to a confirmation bias of pre-existing beliefs. Similarly, political affiliation could affect the likelihood that people mistake real news for fake news. We tested these ideas in two sets of experiments. In the first set, we asked University of Louisiana at Lafayette undergraduates (Experiment 1a $n = 376$) and Mechanical Turk workers in the United States (Experiment 1a $n = 205$; Experiment 1b $n = 201$) to rate how "real" versus "fake" a series of unfamiliar news headlines were. We attributed each headline to one of several news sources of varying political slant. As predicted, we found that source information influenced people's ratings in line with their own political affiliation, although this influence was relatively weak. In the second set, we asked Mechanical Turk workers in the United States (Experiment 2a $n = 300$; Experiment 2b $n = 303$) and University of Louisiana at Lafayette undergraduates (Experiment 2b $n = 182$) to watch a highly publicized "fake news" video involving doctored footage of a journalist. We found that people's political affiliation influenced their beliefs about the event, but the doctored footage itself had only a trivial influence. Taken together, these results suggest that adults across a range of ages rely on information other than news content—such as how they feel about its source—when judging whether news is real or fake. Moreover, our findings help explain how people experiencing the same news content can arrive at vastly different conclusions. Finally, efforts aimed at educating the public in combatting fake news need to consider how political affiliation affects the psychological processes involved in forming beliefs about the news.

**Keywords:** age; confirmation bias; fake news; heuristic approach; politics; source

## 1. Introduction

As the world changes, so too does the way we consume news information. According to survey data, nearly two thirds of people now prefer to read news online [1]. Moreover, those data also indicate that the preference for online news is growing: Of those who say they prefer to watch the news, the proportion who prefer to watch it online almost doubled between 2016 and 2018 [1]. This online access provides people with additional sources of news information and has the potential to widen the scope and quality of the news people encounter. One consequence could be a more well-informed public. However, another

possible consequence is an increase in exposure to "fake news"—a catchall term used by both academics and laypeople to mean content that appears news-like, but varies in how false it is and how harmful it is intended to be [2–4]. This fake news continuum spans (but is not limited to) unprofessional journalism, sponsored content, political propaganda, and wholly fabricated information [2–4]. In the research reported here, we address the general question: How do people across a wide range of ages evaluate the credibility of reported news information?

Presumably, people expect the truth from the news and are motivated to not fall prey to fake news. In other words, people have "accuracy goals" when reasoning about the news [5]. However, news consumers are faced with a difficult challenge because news information is experienced indirectly and therefore carries a degree of ambiguity. How, then, do people determine whether news is true or false? We suspect that people behave as they do under other situations of ambiguity and draw on information beyond the news itself to guide their behavior [6–8].

More specifically, when evaluating whether news is true or false, people could rely only on the central content itself: a headline, the text of an article, or the words spoken by a reporter. However, this task is relatively effortful, particularly under conditions of ambiguity. An easier alternative is to draw on more peripheral cues to help guide evaluations of the news [8]. We therefore predict that people will try to answer the difficult question of whether news is "real" by asking themselves an easier question: Is the source of that news credible [8–10]? In attempting to answer that easier question, people can adopt a less effortful thinking style and draw on their feelings about the source as well as any pre-existing beliefs [6,11–13]. When adopting such a thinking style, people may be swayed by "directional goals," unwittingly relying on evaluative strategies that lead to a desired conclusion but away from the goal of not falling prey to fake news [5].

In this paper, we focus on one piece of information that may play a role in people's evaluations of the news: political affiliation. Evidence now suggests that political affiliation plays a role in determining which news sources people believe produce real or fake news. Specifically, people in the United States who lean politically left rate news sources favored by people who lean politically right more as sources of fake news. The reverse is also true, with people who lean politically right rating news sources favored by people who lean politically left more as sources of fake news [14,15]. Consistent with these findings, survey data show that people are becoming increasingly selective about their media exposure, narrowing down to sources that match their ideology [16]. This trend is especially problematic when politicians actively disseminate falsehoods, for example, [17], and is worsened when news sources adopt poor journalistic standards and misinform the public. In fact, people are increasingly concerned about the integrity of news sources: Public distrust in the media is growing [18–20]. Taken together, this literature is informative about how people choose and evaluate news sources, but is less informative about how people evaluate the news content itself.

What we do not know, then, is the extent to which people's beliefs about news sources affect evaluations of the news. If source credibility acts as a guide, then source information may have consequences for how people interpret and remember the news. Consistent with this idea, we already know that trusted stimuli—like photographs—can change how people remember the news [21,22]. We also know that more credible sources of information generally produce more attitude change, for a review, see [23], and that more credible sources of misinformation are more misleading [24,25]. However, other factors can sometimes change these relationships. Older adults, for example, are more easily misled than younger adults [26,27].

Source information could convey the credibility of reported news, but where does that credibility come from? It may be the product of a number of underlying factors. For example, source credibility could be due to people's beliefs about journalistic integrity, including what standards or processes should be required to declare a piece of information "true" [18–20,28]. Credibility could also be due to the extent to which a news source aligns

with a person's political views [14]. It could also arise as the product of confirmation bias: the extent to which the news content is consistent with and therefore reinforces pre-existing beliefs [12,13].

In an effort to understand what people believe about news information from various sources, we ran two initial experiments (1a and 1b) in which we asked people across a wide range of ages to rate how real (versus fake) they believed a series of news headlines were, varying the ostensible source of those headlines. From related work, we know that political affiliation predicts which sources people believe are credible and that analytical thinking predicts the ability to discern real headlines from fake ones [14,29]. We therefore hypothesized that source information can act as a heuristic cue that people turn to when faced with the difficult task of determining whether headlines are real or fake news. We predicted that people would rate headlines from sources favored by their political affiliation as "real news" more than they would headlines from other sources.

We then followed up on these initial experiments with two additional experiments (2a and 2b) in which we investigated the extent to which people's interpretations of a more familiar and real-world "fake news" event would be affected by the source of information about that event. We predicted that people's interpretations of the event would differ depending on their political views and which version of the event—real or fake—they were exposed to. Across both sets of experiments, we also examined the influence of age in additional exploratory analyses.

## 2. Experiment 1a

The preregistration for this experiment is available at https://aspredicted.org/zs5kv.pdf (accessed on 27 September 2021). The data were collected between 31 January and 21 April 2018.

### 2.1. Method

#### 2.1.1. Subjects

Across all experiments, we aimed to recruit as many subjects as possible, based on subject pool and funding availability. No subject participated in more than one experiment. This goal resulted in a sample size of complete responses from 581 subjects for this experiment, comprised of 376 undergraduate students at the University of Louisiana at Lafayette and 205 Mechanical Turk workers based in the U.S. (381 women, 197 men, 3 unspecified; $M_{age}$ = 27 years, age range: 18–82 years), well above our preregistered minimum sample size. According to a sensitivity analysis, this sample size gives us adequate power to detect a small interaction effect by conventional standards.

#### 2.1.2. Design

We manipulated News Source within subject, attributing headlines to one of five sources. In addition, subjects assigned themselves into one of three Political Affiliation categories.

#### 2.1.3. Materials and Procedure

As a cover story, we told subjects the study was examining visual and verbal learning styles. Then, we presented subjects with 50 news headlines, one at a time, in a randomized order. Each headline was attributed to one of five news sources. Specifically, above each news headline, subjects read "X reported that . . . ," where X was replaced with a news source and the ellipsis was followed by a headline (e.g., "The New York Times reported that . . . Rarely Used Social Security Loopholes, Worth Thousands of Dollars, Closed"). We asked subjects to rate each headline for the extent to which they believed that news story was real news or fake news (1 = Definitely fake news, 5 = Definitely real news).

**News Sources.** We chose the news sources as follows. We gathered an initial list of 42 sources from a study investigating people's beliefs about the prevalence of fake news in various media agencies [14]. For the current experiment, we narrowed this list down to the following four sources: The New York Times, Fox News, Occupy Democrats, and

Breitbart. We chose these sources on face value, in an effort to cover both relatively left- and right-leaning media sources, as well as relatively well-established and new media sources of varying levels of reputed journalistic integrity. We also included an additional unspecified fifth source, achieved by replacing X with the words "It was."

**Headlines.** We constructed the list of news headlines as follows. First, we scoured various U.S. national and international news websites for headlines from the 2015–2016 period. We selected headlines on the basis that they should cover a wide range of topics—including non-political or non-partisan issues—and should make a claim, rather than merely stating an opinion. This initial search produced 167 candidate headlines. We then asked a separate sample of 243 undergraduate students to rate, in a randomized order, the familiarity of each headline (1 = Definitely never seen before, 5 = Definitely seen before). Using these data, we selected a final set of 50 unique, specific headlines that were rated relatively low in familiarity (*M* = 2.32, Range = 1.75–3.43). The final list of headlines is available at https://osf.io/h6qen/ (accessed on 27 September 2021).

No headlines were drawn from our four specified sources. We counterbalanced presentation of the materials such that each subject observed 10 headlines attributed to each source, and each headline was attributed to each source equally often across subjects. We included two attention check items among the headlines that looked similar but specified the response subjects should select if they were paying attention.

Following the headline rating task, we asked subjects how they identified politically (1 = Very conservative, 5 = Very liberal), which political party they were a member of (1 = Democratic party, 2 = Republican party, 3 = Other or none), and basic demographic information. We also administered several exploratory measures: subjects completed the Social Dominance Orientation scale [30], rated how familiar they were with each news source (1 = Not at all familiar, 5 = Extremely familiar), rated how much the source information affected their ratings (1 = Not at all, 5 = A great deal), answered two open-ended questions about the purpose of the study, and indicated if they had looked up any of the headlines. We do not report results from most of these exploratory measures, but the data are available at Fake News - Headlines and Acosta. Available online: https://osf.io/h6qen/ (accessed on 27 September 2021).

*2.2. Results and Discussion*

For all experiments in this article, we report the results of analyses that met the standard criterion of statistical significance (i.e., *p* <0.05). For the interested reader, additional reporting of results can be found in the Supplementary Material.

We only analyzed data from subjects who gave complete responses, and we did not exclude subjects on any other basis, contrary to our preregistration. Most subjects responded correctly to each attention check item (85% and 87%, respectively) and did not look up any headlines (93%). We also deviated from our preregistration in how we created the three political affiliation groups for analysis: Rather than categorizing subjects based on their rated political leaning, we simply used subjects' reported party membership (but using the preregistered groupings leads to similar results and conclusions; see Supplementary Material).

Of the 581 subjects, 229 identified as Republicans, 177 as Democrats, and 175 as Other (or none). Distributions of the political leaning variable were consistent with these data: The modal selections were "somewhat conservative" for Republicans, "somewhat liberal" for Democrats, and "Moderate" for Other.

Recall that our primary question was: To what extent does political affiliation influence how source information affects people's interpretations of the news? To answer that question, we examined subjects' mean headline ratings as a function of their political affiliation and news source. Table 1 shows the mean rating for each condition. A Repeated Measures Analysis of Variance (RM-ANOVA) on these ratings revealed a statistically significant interaction between political affiliation and news source, suggesting that the influence of political affiliation on headline ratings depends on source information, *F*(8,

2312) = 3.09, $p < 0.01$, $\eta^2_p = 0.011$. We also included age as a covariate in an additional exploratory Repeated Measures Analysis of Covariance (RM-ANCOVA), and found only a main effect of Age, such that each year of aging was associated with a small shift toward rating headlines more as real news, irrespective of source or political affiliation, $B = 0.005$, $t(579) = 3.77$, $p < 0.01$.

**Table 1.** Descriptive Statistics for Ratings of News Classified by Source of Material and Subjects' Political Affiliation.

| Source [a] | Political Affiliation [b] | | | | | | | |
| | Republican | | Democrat | | Other | | None | |
| | M | 95% CI | M | 95% CI | M | 95% CI | M | 95% CI |
|---|---|---|---|---|---|---|---|---|
| | | | Experiment 1a | | | | | |
| New York Times | 3.30 | [3.22, 3.37] | 3.47 | [3.39, 3.56] | 3.32 | [3.24, 3.40] | – | – |
| Fox News | 3.35 | [3.27, 3.42] | 3.28 | [3.20, 3.36] | 3.26 | [3.19, 3.34] | – | – |
| Occupy Democrats | 3.20 | [3.12, 3.27] | 3.35 | [3.27, 3.43] | 3.21 | [3.14, 3.29] | – | – |
| Breitbart | 3.22 | [3.14, 3.30] | 3.24 | [3.15, 3.33] | 3.16 | [3.08, 3.24] | – | – |
| Unspecified | 3.20 | [3.12, 3.28] | 3.33 | [3.25, 3.40] | 3.19 | [3.11, 3.27] | – | – |
| | | | Experiment 1b | | | | | |
| CNN | 3.30 | [3.11, 3.48] | 3.25 | [3.13, 3.37] | 3.50 | [3.37, 3.63] | – | – |
| Fox News | 3.39 | [3.24, 3.55] | 3.13 | [3.02, 3.24] | 3.44 | [3.31, 3.57] | – | – |
| Unspecified | 3.13 | [2.96, 3.30] | 3.17 | [3.06, 3.28] | 3.34 | [3.24, 3.45] | – | – |
| | | | Experiment 2a | | | | | |
| Altered | 2.65 | [2.37, 2.93] | 2.02 | [1.71, 2.33] | 2.02 | [1.67, 2.37] | – | – |
| Looped | 2.80 | [2.45, 3.15] | 1.72 | [1.48, 1.96] | 2.01 | [1.69, 2.32] | – | – |
| Original | 2.96 | [2.69, 3.23] | 1.92 | [1.65, 2.19] | 2.30 | [1.99, 2.60] | – | – |
| | | | Experiment 2b | | | | | |
| Altered | 2.48 | [2.32, 2.64] | 2.14 | [1.98, 2.30] | 1.98 | [1.50, 2.46] | 2.17 | [1.98, 2.36] |
| Original | 2.69 | [2.53, 2.86] | 2.08 | [1.91, 2.24] | 2.41 | [1.94, 2.88] | 2.36 | [2.20, 2.53] |

Note. In Experiments 1a and 1b, ratings concerned the "realness" of various news headlines; in Experiments 2a and 2b, ratings were a composite of four items concerning the negativity of a CNN journalist's interaction with a White House intern as depicted in video footage recorded during a press conference given by President Trump. [a] In Experiments 1a and 1b, headlines were attributed to various news sources; in Experiments 2a and 2b source was one of several versions of a videoed event. [b] In Experiments 1a, 1b, and 2a, "Other or none" was a single political affiliation response option, whereas in Experiment 2b, "Other" and "None" were distinct response options.

To determine where any meaningful differences occurred, we then ran five one-way ANOVAs testing the influence of political affiliation on mean headline ratings for each news source (we did not explicitly specify these follow-up analyses in our pre-registration). These analyses yielded mixed results. Subjects' political affiliation had no appreciable influence when headlines came from the two sources favored by people who lean politically right (all $p$ values > 0.26). However, subjects' political affiliation did have an influence when headlines came from the remaining three news sources, $F_{\text{New York Times}}(2, 578) = 5.17$, $p < 0.01$, $\eta^2_p = 0.018$; $F_{\text{Occupy Democrats}}(2, 578) = 4.57$, $p = 0.01$, $\eta^2_p = 0.016$; $F_{\text{Unspecified Source}}(2, 578) = 3.34$, $p = 0.04$, $\eta^2_p = 0.011$.

More specifically, Tukey-corrected post hoc comparisons for those three sources revealed that Democrats rated headlines from the New York Times as slightly more real than Republicans ($M_{\text{Diff}} = 0.18$, 95% CI [0.04, 0.31], $p = 0.01$) or Others ($M_{\text{Diff}} = 0.15$, 95% CI [0.01, 0.29], $p = 0.04$). Similarly, Democrats rated headlines from Occupy Democrats as slightly more real than Republicans ($M_{\text{Diff}} = 0.15$, 95% CI [0.03, 0.28], $p = 0.01$) or Others ($M_{\text{Diff}} = 0.14$, 95% CI [0.00, 0.27], $p = 0.04$). Finally, Democrats rated headlines from an unspecified source as more real, in the mean, than Republicans ($M_{\text{Diff}} = 0.12$, 95% CI [−0.01, 0.26], $p = 0.07$) or Others ($M_{\text{Diff}} = 0.14$, 95% CI [−0.00, 0.28], $p = 0.06$). These last differences were not statistically significant once adjusted for multiple comparisons, however.

Taken together, this collection of results is partially consistent with our hypothesis. We predicted that people would rate news headlines from sources favored by their political affiliation as more real than headlines from other sources. That prediction was correct—but only for headlines attributed to sources favored by people who lean politically

left (Democrats). How are we to explain these results? One possibility is that Democrats—and only Democrats—make meaningful distinctions among news sources, but we can think of no theoretical reason this explanation would be true. An alternative possibility is that the sources we used varied in unanticipated ways. In fact, exploratory examination of subjects' source familiarity ratings reveals data consistent with this idea: The New York Times and Fox News were rated more familiar than Occupy Democrats and Breitbart ($M_{\text{New York Times}}$ = 3.36, 95% CI [3.25, 3.47]; $M_{\text{Fox News}}$ = 3.56, 95% CI [3.46, 3.66]; $M_{\text{Occupy Democrats}}$ = 1.69, 95% CI [1.60, 1.78]; $M_{\text{Breitbart}}$ = 1.72, 95% CI [1.62, 1.81]). We also note that the headline rating differences were small, suggesting that our sources may not be construed as meaningfully different from one another in terms of their credibility. We conducted Experiment 1b to address these concerns.

## 3. Experiment 1b

The preregistration for this experiment is available at https://aspredicted.org/pi83g.pdf (accessed on 27 September 2021). The data were collected on 17 February 2019.

### 3.1. Method

#### 3.1.1. Subjects

We collected complete responses from 201 Mechanical Turk workers based in the U.S. (112 women, 88 men, 1 unspecified, $M_{\text{age}}$ = 40 years, age range: 18–77 years), one more than our preregistered sample size. A sensitivity analysis indicates this sample size gives us adequate power to detect a small-to-medium interaction effect by conventional standards.

#### 3.1.2. Design

We manipulated News Source within subject, attributing headlines to one of three sources. In addition, subjects assigned themselves into one of three Political Affiliation categories.

#### 3.1.3. Materials and Procedure

The experiment was identical to Experiment 1a, except as follows.

**News Sources.** We chose different news sources for this experiment. To more formally identify and quantify experimentally useful news sources, we first asked a separate sample of 202 Mechanical Turk workers to provide familiarity, trustworthiness, and bias ratings for each of the original list of 42 news sources [14]. The preregistration for this norming study is available at FA2018 - Fake news - Sources norming (#13611). https://aspredicted.org/4ep7p.pdf (accessed on September 2021). Subjects observed the names of these news sources, one at a time, in a randomized order. For each source, subjects provided a familiarity rating (1 = Not at all, 5 = Extremely), a trustworthiness rating (1 = Not at all, 5 = Extremely), and a bias rating (1 = Strong liberal bias, 5 = Strong conservative bias). We also asked subjects about their own political leaning and party affiliation. From these data, we identified the two news sources rated maximally different on trustworthiness across Democrats and Republicans: CNN ($M_{\text{Democrat}}$ = 3.73, $M_{\text{Republican}}$ = 2.52) and Fox News ($M_{\text{Democrat}}$ = 1.95, $M_{\text{Republican}}$ = 3.27).

Subjects in the current study were presented with 48 headlines, randomly selected from the original set of 50, so that subjects rated 16 headlines per source. Each headline was attributed to one of three news sources. Specifically, subjects read: "X reported that . . . ," where X was replaced with either "CNN," "Fox News," or "It was" for the unspecified source. This time, subjects did not rate the familiarity of each source. The data are available at https://osf.io/h6qen/ (accessed on 27 September 2021).

### 3.2. Results and Discussion

We analyzed data only from subjects who gave complete responses, and we did not exclude subjects on any other basis, contrary to our preregistration. Most subjects responded correctly to each attention check item (97% and 98%, respectively) and did not look up any headlines (98%).

Of the 201 subjects, 44 identified as Republicans, 92 as Democrats, and 65 as Other (or none). Distributions of the political leaning variable were consistent with these data: The modal selections were "somewhat conservative" for Republicans, "somewhat liberal" for Democrats, and "Moderate" for Other.

Recall that our primary question, as in Experiment 1a, was: To what extent does political affiliation influence how source information affects people's interpretations of the news? To answer that question, we examined subjects' mean headline ratings as a function of their political affiliation and news source. Table 1 shows the mean rating for each condition. A RM-ANOVA on mean headline ratings revealed—as in Experiment 1a—a statistically significant interaction between political affiliation and news source, suggesting that the influence of political affiliation on headline ratings depends on source information, $F(4, 396) = 2.52$, $p = 0.04$, $\eta^2_p = 0.025$. We also included age as a covariate in an additional exploratory RM-ANCOVA, but found that age had no meaningful influence (all age-related $p$ values > 0.18).

To determine where any meaningful differences occurred, we then ran three one-way ANOVAs testing the influence of political affiliation on mean headline ratings for each news source (we did not explicitly specify these follow-up analyses in our preregistration). As in Experiment 1a, these analyses yielded mixed results: Subjects' political affiliation influenced ratings of headlines only from CNN and Fox News, $F_{CNN}(2, 198) = 3.84$, $p = 0.02$, $\eta^2_p = 0.037$; $F_{Fox\ News}(2, 198) = 7.78$, $p < 0.01$, $\eta^2_p = 0.073$.

More specifically, Tukey-corrected post hoc comparisons for these two sources revealed that Democrats rated headlines from CNN as less real than Others ($M_{Diff} = 0.25$, 95% CI [0.03, 0.47], $p = 0.02$). Democrats also rated headlines from Fox News as less real than both Republicans ($M_{Diff} = 0.26$, 95% CI [0.04, 0.49], $p = 0.02$) and Others ($M_{Diff} = 0.31$, 95% CI [0.11, 0.51], $p < 0.01$).

Taken together, this collection of results is consistent with our hypothesis, but only partially so. We predicted that people would rate headlines attributed to sources favoring their political affiliation as more real than headlines attributed to other sources. That prediction was correct, but in contrast to Experiment 1a, only for headlines attributed to a source favoring people who lean politically right: Fox News.

Overall, the results of Experiments 1a and 1b suggest that source information contributes to people's interpretations of the news. However, there are two key limitations to this conclusion. First, the observed differences were small, and not entirely consistent across our two samples. Consider, however, that subjects were provided with only the mere name of a source. It is perhaps surprising that such limited information can have any influence at all. Second, the headlines were normed to be relatively unfamiliar. We chose to use unfamiliar headlines in an effort to control for pre-existing knowledge, but it is possible that unfamiliar headlines convey so little information that they are almost meaningless. Again, however, it may be surprising that source information can influence interpretations of almost meaningless headlines.

Having conducted these initial investigations, we were then presented with a unique opportunity. In November of 2018, a United States White House intern attempted to take a microphone away from CNN's Jim Acosta during a press conference. Acosta clung to the microphone, resulting in brief contact between the two. Shortly afterward, then-Press Secretary Sarah Huckabee Sanders posted video footage of the interaction to Twitter. Sanders used the video as justification for revoking Acosta's White House press pass, claiming his behavior was inappropriate. However, rather than posting the original CSPAN footage, Sanders posted a subtly altered video that appears to have originated from a conservative media site [31].

Several media agencies raised concerns about the potential suggestive influence of this manipulated footage. Consistent with these concerns, a partisan split emerged, with those on the left tending to claim Acosta's behavior was unremarkable, while those on the right tended to claim his behavior was problematic. One explanation for this split is that the version of the video people observed guided their interpretations of Acosta's behavior.

However, we suspected that the explanation was more nuanced, hypothesizing that any influence of the video would depend on political affiliation. More specifically, we predicted that due to beliefs about media sources, Republicans would be more susceptible to any potential influence of the altered video than Democrats. In Experiments 2a and 2b, we therefore tested the extent to which altered video footage of a real-world event affected people's interpretations of that event. In contrast to Experiments 1a and 1b, video footage of a real-world event provides a richer context than a sparse headline and allows us to explore the role of familiarity with the news story.

## 4. Experiment 2a

The preregistration for this experiment is available at https://aspredicted.org/da3hg.pdf (accessed on 27 September 2021). The data were collected on 2 May 2019.

### *4.1. Method*

#### 4.1.1. Subjects

We collected complete responses from 300 Mechanical Turk workers based in the U.S. (200 women, 98 men, 2 unspecified, $M_{age}$ = 38 years, age range: 18–76 years). In a deviation from our preregistration, we did not recruit additional subjects from an undergraduate population and so we collected more responses than planned from Mechanical Turk. A sensitivity analysis indicates this sample size gives us adequate power to detect a small-to-medium interaction effect by conventional standards.

#### 4.1.2. Design

We manipulated Video Version between subjects, showing each subject one of three versions of the event. In addition, subjects assigned themselves into one of three Political Affiliation categories.

#### 4.1.3. Materials and Procedure

As a cover story, we told subjects the study was examining visual and verbal learning styles. Then, we asked subjects to watch a brief video of an interaction between a journalist and a White House intern during a press conference, randomly assigning them to see one of three versions of this event. We collected these data approximately six months after the event occurred. Subjects then made several ratings related to the depicted interaction, to gauge how they interpreted the journalist's behavior.

**Video Versions.** The video versions were as follows. The "altered" version of the video is that tweeted by the then-Press Secretary. It is a 15 s clip with no audio that loops the brief interaction between the journalist and the White House intern a total of six times; on the second loop, and again on the third loop, the video zooms in and it remains zoomed in thereafter. From the original CSPAN footage, we created two additional versions, each 15 s long with the audio removed. Our "looped" version of the video consists of the brief interaction, looped. The "original" version of the video consists of the interaction itself, as well as approximately 6 s of footage preceding the interaction and 6 s of footage following it. Links to all three videos are available at https://osf.io/h6qen/ (accessed on 27 September 2021).

**Ratings Items.** Subjects made four key ratings related to the interaction they had just seen. Specifically, subjects first rated the harmfulness of the journalist's behavior toward the intern (1 = entirely harmless, 4 = entirely harmful), then the reasonableness of the journalist's behavior toward the intern (1 = entirely unreasonable, 4 = entirely reasonable; reverse scored). Next, we told subjects that as a result of the interaction, the White House took away the journalist's press pass, meaning he was banned from the White House. Subjects then rated the White House's response (1 = entirely unreasonable, 4 = entirely reasonable). We then told subjects that a federal judge later ruled that taking away the journalist's press pass was a violation of his right to a fair and transparent process, ordering that the ban be lifted. Subjects then rated the judge's ruling (1 = entirely unreasonable, 4 = entirely reasonable; reverse scored).

Following these ratings, subjects provided information about their political affiliation and basic demographics, as in the previous experiments. We also administered several exploratory measures, asking subjects to rate how familiar they were with the events shown in the video, prior to the study (1 = entirely unfamiliar, 4 = entirely familiar), as well as questions variously addressing their prior familiarity with specific pieces of information related to the event and its aftermath, characteristics of the video version they observed, if they had looked up any related information during the study, and the purpose of the study; we do not report results for most of these measures here. The data are available at https://osf.io/h6qen/ (accessed on 27 September 2021).

### 4.2. Results and Discussion

We analyzed only the data from subjects who gave complete responses, and we did not exclude subjects on any other basis, contrary to our preregistration. Most subjects did not look up any related information (97%).

Of the 300 subjects, 80 identified as Republicans, 133 as Democrats, and 87 as Other (or none). Distributions of the political leaning variable were consistent with these data: The modal selection was "somewhat conservative" for Republicans, "somewhat liberal" for Democrats, and "Moderate" for Other.

Recall that our primary question was: To what extent does political affiliation influence how people interpret video footage of a real-world news event? To answer that question, we first calculated, for each subject, an average of their ratings across the four key items. We preregistered to conduct multivariate analyses across these four ratings, but because they were highly correlated ($r$s = 0.58–0.69; Cronbach's $\alpha$ = 0.87) we chose instead to combine them for univariate analysis (but conducting the preregistered analyses leads to similar results and conclusions; see Supplementary Material). Higher scores on this composite measure reflect more negative interpretations of the journalist's behavior. Table 1 shows the mean composite rating for each condition.

We then examined subjects' composite rating as a function of the video version they observed and their political affiliation. A two-way ANOVA revealed only a main effect of political affiliation, suggesting that when it came to how negatively people interpreted the journalist's behavior, only political affiliation mattered $F(2, 291) = 28.95$, $p < 0.01$, $\eta^2_p = 0.166$. More specifically, Tukey-corrected post hoc comparisons revealed that Republicans rated the journalist's behavior more negatively than did Democrats ($M_{Diff} = 0.92$, 95% CI [0.64, 1.21], $p < 0.01$) and Others ($M_{Diff} = 0.68$, 95% CI [0.37, 0.99], $p < 0.01$).

We also included age as a covariate in an additional exploratory RM-ANCOVA and found that each year of aging was associated with a shift in judgment about the journalist's behavior, but the direction and strength of this shift depended on political affiliation, $F_{Age \times Political\ Affiliation}(2, 292) = 4.48$, $p = 0.01$, $\eta^2_p = 0.030$. More specifically, for Democrats only, each year of aging was associated with a statistically significant shift toward a more positive interpretation of the journalist's behavior, $B = -0.015$, $t(130) = 2.34$, $p = 0.02$.

These findings indicate that concerns over the suggestive nature of the altered video may have been unwarranted. What, then—if not the video—drives the observed differences across the political spectrum? One possibility is prior familiarity with the event itself. To explore this possibility, we split subjects into two groups classified according to their ratings of prior familiarity with the event: Subjects who reported they were entirely or somewhat unfamiliar with the event were classified as "unfamiliar" ($n = 146$), while subjects who reported they were somewhat or entirely familiar with the event were classified as "familiar" ($n = 154$). We then re-ran the two-way ANOVA for each of these groups in turn. Although exploratory, the results suggest that familiarity mattered: The only statistically significant factor was political affiliation—with the same pattern of means as above—and only among those who were already familiar with the event (Familiar: $p < 0.01$; Unfamiliar: $p = 0.08$).

We conducted Experiment 2b to replicate these findings with a simplified design and a slightly larger sample.

## 5. Experiment 2b

The preregistration for this experiment is available at https://aspredicted.org/437a8 .pdf (accessed on 27 September 2021). The data were collected between 18 September and 21 November 2019.

### 5.1. Method

5.1.1. Subjects

We collected complete responses from a total of 485 subjects, comprised of 182 undergraduate students at the University of Louisiana at Lafayette and 303 Mechanical Turk workers based in the U.S. (292 women, 186 men, 7 unspecified, $M_{age}$ = 32 years, age range: 18–75 years), in line with our preregistered sampling plan. A sensitivity analysis indicates this sample size gives us adequate power to detect a small-to-medium interaction effect by conventional standards.

5.1.2. Design

We manipulated Video Version between subjects, showing each subject one of two versions of the event. In addition, subjects assigned themselves into one of four Political Affiliation categories.

5.1.3. Materials and Procedure

The experiment was identical to Experiment 2a, except as follows. We collected these data approximately 10 months after the event occurred. Because we found no effects of video version in Experiment 2a, we simplified the design, dropping the "looped" version of the video and randomly assigning subjects to watch either the "altered" version or the "original" version. We also allowed subjects to differentiate between having an "Other" political affiliation and "None." Finally, we included some slightly different exploratory measures, which we do not report the results of here. The data are available at https: //osf.io/h6qen/ (accessed on 27 September 2021).

### 5.2. Results and Discussion

We analyzed data only from subjects who gave complete responses, and we did not exclude subjects on any other basis, contrary to our preregistration. Most Mechanical Turk workers did not look up any related information (95%).

Of the 485 subjects, 130 identified as Republicans, 184 as Democrats, 143 as None, and 28 as Other. Distributions of the political leaning variable were consistent with these reports: The modal selections were "somewhat conservative" for Republicans, "somewhat liberal" for Democrats, and "Moderate" for Other and None.

Recall that our primary question was: To what extent does political affiliation influence how people interpret video footage of a real-world news event? To answer that question, we again calculated, for each subject, an average of their ratings across the four key items. As before, we preregistered to conduct multivariate analyses across these four ratings, but because they were all at least moderately correlated ($r$s = 0.40–0.61; Cronbach's $\alpha$ = 0.80) we chose instead to combine them for univariate analysis (but conducting the preregistered analyses leads to similar results and conclusions; see Supplementary Material). Table 1 shows the mean composite rating for each condition.

We examined subjects' composite rating as a function of the video version they observed and their political affiliation. A two-way ANOVA revealed main effects of video version, $F(1, 477)$ = 4.78, $p$ = 0.03, $\eta^2_p$ = 0.010, and political affiliation, $F(3, 477)$ = 10.77, $p < 0.01$, $\eta^2_p$ = 0.063. These results suggest that the version of the event people observed and their political affiliation each mattered for how they interpreted the journalist's behavior.

More specifically—and contrary to our predictions—people who viewed the "original" version of the video gave slightly more negative ratings of the journalist's behavior than people who viewed the "altered" version ($M_{Diff}$ = 0.14, 95% CI [−0.00, 0.27]). Tukey-corrected post hoc comparisons further revealed that, in terms of people's political affiliation, Repub-

licans rated the journalist's behavior more negatively than Democrats ($M_{\text{Diff}}$ = 0.49, 95% CI [0.27, 0.71], $p < 0.01$), Others ($M_{\text{Diff}}$ = 0.40, 95% CI [0.00, 0.80], $p = 0.05$), and members of no party ($M_{\text{Diff}}$ = 0.32, 95% CI [0.09, 0.56], $p < 0.01$).

We also included age as a covariate in an additional exploratory ANCOVA and found that each year of aging was associated with a shift in interpretation of the journalist's behavior, but the direction and strength of this shift depended on political affiliation, $F_{\text{Age x Political Affiliation}}(3, 476) = 3.67$, $p = 0.01$, $\eta^2_{\text{p}} = 0.023$. More specifically, for Democrats and those reporting not belonging to any political party, each year of aging was associated with a statistically significant shift toward a more positive rating of the journalist's behavior, $B_{\text{Democrats}} = -0.017$, $t(181) = 4.33$, $p < 0.01$; $B_{\text{None}} = -0.012$, $t(141) = 2.54$, $p = 0.01$.

This pattern of results is largely consistent with the findings of Experiment 2a and reinforces the idea that concerns over the suggestive nature of the altered video may have been unwarranted. As in Experiment 2a, we wondered about the influence of subjects' prior familiarity. We again split subjects into two groups, classifying them as "unfamiliar" ($n = 309$) or "familiar" ($n = 178$) with the event, according to their rating of prior familiarity. We then re-ran the two-way ANOVA for each of these groups in turn. In these exploratory analyses, only political affiliation remained statistically significant—with the same patterns of means as above—and only for those who indicated prior familiarity (Familiar: $p < 0.01$; Unfamiliar: $p = 0.09$). These results are consistent with the findings of Experiment 2a, suggesting again that familiarity with the event matters.

## 6. General Discussion

Across four experiments encompassing a variety of news sources and a real-world event that varied in familiarity, we found that the influence of source depends on political beliefs.

In Experiment 1a, we found that Democrats rated unfamiliar news headlines as more likely to be real than Republicans or Others did—but only when those headlines were attributed to news sources favored by Democrats. This result shows that a simple change to the ostensible source of news information can affect people's interpretations of that news. In addition, we found that the older people were, the more "real" they rated headlines, regardless of the source of those headlines or people's political affiliation. Our sources had not been normed for credibility, however, leaving room for alternative interpretations. In Experiment 1b, we sought to resolve this issue and build on our initial findings, examining two news sources previously rated most distinct in trustworthiness across the political spectrum. Here, we found that Democrats rated unfamiliar headlines as less likely to be real than Republicans or Others—but only when those headlines were attributed to a news source not favored by Democrats.

In Experiments 2a and 2b, we found evidence to suggest that prior knowledge of a real-world "fake news" event strongly influences people's beliefs about that event. More specifically, when people indicated they already knew about the depicted event—that is, the interaction between CNN's Jim Acosta and a White House intern—ratings about the journalist's behavior were consistent with political affiliation: Democrats rated the journalist's behavior more favorably than Republicans. Moreover, this influence of familiarity dwarfed any influence of the version of the video people observed. We also found that the older people were, the more positively they rated the journalist's behavior, but only among Democrats or people who belonged to no political party. These results suggest that concerns about the suggestive nature of the altered video may have been unwarranted, especially when considering that those unfamiliar with the event rated the journalist's behavior *more* favorably after watching the altered video than after watching the original CSPAN footage.

Our findings are consistent with related work showing that people's political beliefs predict which news sources they consider to be "fake news" [14]. Our data build on this work, suggesting that in some cases, differences in beliefs about the trustworthiness of news sources carries forward into judgments of the veracity of news information. That finding is concerning, because related research shows that "fake news" is often political in

nature and can have serious consequences, such as non-compliance with behaviors that inhibit the spread of a deadly virus [32–34]. Our research is also reminiscent of other work showing that individual differences—like age, the need to see the world as structured, or the propensity to think analytically—predict endorsement of or skepticism about "fake news" and misinformation [26,27,29,35,36]. With respect to age specifically, we found two small but noteworthy patterns. First, age was positively associated with the belief that news headlines were "real" in Experiment 1a. This finding should be interpreted cautiously, however, because we did not observe the same association in Experiment 1b. Second, age was positively associated with more favorable views of the journalist's behavior in Experiments 2a and 2b—although not for Republicans. Together, these findings are consistent with work showing differences in the ability to think critically as people age [37]. Finally, our results also dovetail with prior research demonstrating that people are more easily misled by sources of information deemed credible [24,25].

One limitation, however, is that news source information appears to have only a small influence on people's beliefs about the news. Take, for example, the finding from Experiment 1b, in which Democrats rated headlines attributed to Fox News as less real than either Republicans or Others. The confidence intervals for those differences ranged from 0.04 to 0.51—or put another way, from almost zero to half of a point along a 5-point scale. However, considering that subjects were given sparse information in the form of brief and unfamiliar news headlines, any effect at all may seem surprising.

There are at least three possible explanations for the small size of these effects. The first is that people require more context (e.g., a longer news article) for news source information to powerfully sway interpretations of the news. The results from Experiments 2a and 2b are consistent with this idea, because differences in event interpretations due to political affiliation were strongest amongst those already familiar with the event. The second explanation is that people do not rely on source information when evaluating news content that is already relatively plausible [38–40]. The third explanation—and one we should take seriously when designing interventions to help people detect fake news—is that people are increasingly skeptical of news sources in general [18–20]. If that trend continues, then it will become difficult to find any meaningful differences in people's interpretations of the news according to where that news is sourced, because all sources will eventually be considered "fake news." In fact, given the proliferation of digitally altered footage in which people are convincingly replaced with others (i.e., "deep fakes"), we may be approaching a tipping point, beyond which no news will be considered credible [41].

Another limitation is that we lacked control over what people already knew about the real-world "fake news" event in Experiments 2a and 2b, instead choosing to measure naturally occurring familiarity. We therefore cannot be sure what caused differences in interpretations of the event amongst those already familiar with the event. We know that the video itself is an inadequate explanation, because video version had no meaningful influence among people who were unfamiliar with the event. We suspect a likely explanation is that Democrats and Republicans encountered different reports of the event due to selective news source consumption [16]. Consistent with this explanation, Fox News's reporting of the event featured a Tweet from a conservative commentator stating that Acosta had bullied the intern and should have his press credentials revoked [42].

One implication of this research hinges on the finding that the same news was interpreted differently when it came from different sources. That finding implies that people rely on more than just the news content when forming beliefs about the news. This implication is consistent with other work showing that people sometimes draw on whatever is available—like how easy it feels to process information—when making judgments about various targets [6,8,43]. It is similarly consistent with an explanation in which people's political motivations influence their reasoning about the news, and more generally with work showing that people find information more persuasive when it comes from a more credible source [5,23,44]. Finally, it is consistent with a framework in which people use source information when making attributions about remembered details [10]. A future

study could examine the extent to which people can remember the source of encountered news information. We suspect that given the trend towards news source selectivity, people will be relatively good at remembering those sources they are familiar with, but relatively poor at remembering those sources they are not familiar with [16].

This narrowing source selectivity likely acts as a negative feedback loop, serving to reinforce pre-existing, ideologically aligned beliefs—even when those beliefs are not accurate [5]. Moreover, people may be unaware such selectivity is happening: Multiple technology giants such as Google and Facebook curate content according to algorithms, resulting in externally generated selectivity [45]. Such a "filter bubble" may be especially concerning when news sources blatantly misinform. Take the recent example of Fox News publishing digitally altered images, placing an armed guard into photos of protests in Seattle [46].

What steps could be taken to reverse this selectivity? Can we successfully encourage people to engage with a wider variety of news sources and to be more critical of news reporting? Some efforts are underway, though it remains to be seen whether these approaches are successful [47–49]. Given the increasing distrust in the media, a more successful approach may be to make systemic regulatory changes to the media itself [18–20]. One idea, for example, is to re-establish the Fairness Doctrine, ensuring that broadcasters cover multiple aspects of controversial issues [50]. Such regulatory measures may ultimately increase accurate and decrease inaccurate news reporting, and in doing so reduce the burden on individuals to detect fake news.

**Supplementary Materials:** The following are available online at https://www.mdpi.com/article/10.3390/soc11040119/s1, Additional Analyses.

**Author Contributions:** Conceptualization, R.B.M. and M.S.; methodology, R.B.M. and M.S.; validation, R.B.M. and M.S.; formal analysis, R.B.M. and M.S.; investigation, R.B.M.; resources, R.B.M.; data curation, R.B.M.; writing—original draft preparation, R.B.M.; writing—review and editing, R.B.M. and M.S.; visualization, R.B.M. and M.S.; supervision, R.B.M. and M.S.; project administration, R.B.M.; funding acquisition, R.B.M. All authors have read and agreed to the published version of the manuscript.

**Funding:** This research received no external funding.

**Institutional Review Board Statement:** The study was conducted according to the guidelines of the Declaration of Helsinki, and approved by the Institutional Review Board of University of Louisiana at Lafayette (FA17-23PSYC, approved 12 September 2017).

**Informed Consent Statement:** Informed consent was obtained from all subjects involved in the study.

**Data Availability Statement:** The data presented in this study are openly available in Open Science Framework at https://osf.io/h6qen/ accessed on 27 September 2021.

**Conflicts of Interest:** The authors declare no conflict of interest.

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
