# Peer review of "Source Information Affects Interpretations of the News across Multiple Age Groups in the United States"

_societies, doi:10.3390/soc11040119_

Round 1

Reviewer 1 Report

Line

Remark

63

Replace “Here,” with “In this paper,”

23

“[redacted]”  Is this meant this way?

118

See above “[redacted]”

This paper analyzes the credibility of news and the ability of news consumers to differentiate between "real" and "fake" news. During several experiments, the authors could gather data which confirmed that political affiliation/orientation of respondents is positively correlated with their faith in specific representatives of mass media. At the same time, the respondents' willingness to allow for "altered videos" respectively to give more credit to specific news institutions in accordance with their demographic group appeared both intuitive and easy to manipulate.

This paper belongs to the category of "soft(er) research" which focuses on contents to be conveyed and less on the strictness of the methodological procedure itself. Nevertheless, it is an empirically founded paper which will bring necessary insights into the functioning of the complex relationship between distributors and consumers of news in mass-media.

Reviewer 2 Report

This article presents a very interesting experiment, the methodology is correct and the results appear sound and promising, so I have nothing to say about this part.

Nonetheless, the article LACKS A THEORETICAL FRAMEWORK.

Therefore, although promising, this work cannot be published without a proper theoretical framework that presents and frame the research question.

1) This framework should at least contain a definition of what the author(s) call “fake news” and an overview of the broader topic of disinformation, addressing the different definitions of misinformation, disinformation and malinformation.

There is a lot of literature that should be cited.

For a full discussion, see  for example the Unesco’s handbook (https://en.unesco.org/fightfakenews)

On definitions:

Edson C. Tandoc Jr., Ryan J. Thomas, Lauren Bishop, What Is (Fake) News? Analyzing News Values (and More) in Fake Stories, https://doi.org/10.17645/mac.v9i1.3331

2) This framework should contextualize disinformation in the broader context of post-truth politics and the evolution of political discourse.

Some references:

Silvio Waisbord (2018) Truth is What Happens to News, Journalism Studies, 19:13, 1866-1878, DOI: 10.1080/1461670X.2018.1492881

Cervi, L., & Carrillo-Andrade, A. (2019). Post-truth and disinformation: Using discourse analysis to understand the creation of emotional and rival narratives in Brexit. ComHumanitas: Revista Científica De Comunicación, 10(2), 125-149. https://doi.org/10.31207/rch.v10i2.207

Cervi, L.; García, F.; Marín-Lladó, C. Populism, Twitter, and COVID-19: Narrative, Fantasies, and Desires. Soc. Sci. 2021, 10, 294. https://doi.org/10.3390/socsci10080294

In addition, the conclusion should be able to show what this experiment adds to the existing literature and compare it with similar experiments done in other contexts/using other methodologies.

I am sure that with a proper theoretical framework, this work will make an interesting contribution.

Round 2

Reviewer 2 Report

The article has improved and it is now publishable

Author Response

Many thanks.